# Round goby (*Neogobius melanostomus*) $\delta^{13}$C/$\delta^{15}$N discrimination values and comparisons of diets from gut content and stable isotopes in Oneida Lake

**Anna M. Poslednik**¤*, **Thomas M. Evans**, **James R. Jackson**, **Anthony J. VanDeValk**, **Thomas E. Brooking**, **Lars G. Rudstam**

Department of Natural Resources and the Environment, Cornell Biological Field Station, Cornell University, Bridgeport, New York, United States of America

¤ Current address: Virginia Institute of Marine Science, College of William & Mary, Gloucester Point, Virginia, United States of America

* amp438@cornell.edu

## Abstract

Gut content analyses have found that round gobies (*Neogobius melanostomus*) are highly dependent on dreissenid mussels but stable isotope analysis has often suggested that the dreissenid contribution is lower. However, estimation of dietary contributions with stable isotopes relies on accurate discrimination factors (fractionation factors). To test if discrimination values commonly used in aquatic food web studies are suitable for round gobies, we collected round gobies from Oneida Lake, raised them for 63 days under four different diets (*Chironomus plumosus*, *Mytilus chilensis*, *Dreissenia* spp., *Euphausia superba*) and measured the change in white muscle $\delta^{13}$C and $\delta^{15}$N. Gobies were also collected throughout Oneida Lake for gut content and stable isotope analysis. Diets changed as round gobies grew, with small round gobies (17-42mm) feeding mostly on cladocera and chironomids, intermediate sized gobies (43-94mm) transitioning from chironomid to dreissenid consumption, and larger gobies (95-120mm) predominantly consuming dreissenids, similar to findings in other studies. Discrimination factors were obtained by fitting a commonly used asymptotic regression equation describing changes in fish $\delta^{13}$C and $\delta^{15}$N as a function of time and diet stable isotope ratios. The discrimination factor determined for $\delta^{13}$C (-0.4‰ ± 0.32, SE) was lower than the "standard" value of 0.4‰, while that of $\delta^{15}$N (4.0‰ ± 0.32, SE) was higher than the standard value of 3.4‰. Turnover rates for both $\delta^{13}$C and $\delta^{15}$N were estimated as 0.02 ‰*day$^{-1}$. The use of experimentally determined discrimination factors rather than "standard" values resulted in model estimates that agree more closely with the observed increasing importance of dreissenids in gut content of larger gobies. Our results suggest that the importance of dreissenid mussels inferred from stable isotope studies may be underestimated when using standard isotopic discrimination values.

**Data Availability Statement:** All relevant data are within the paper and its Supporting Information files.

**Funding:** This research was supported by Cornell University, New York State Department of Environmental Conservation (NYSDEC) grant F-63-R to JRJ, and the John and Janet Forney endowment. https://www.dec.ny.gov/outdoor/41034.html https://cbfs.dnr.cornell.edu/internship-programs/ The funders had no role in study design, data collection and analysis, decision to publish, or preparation of the manuscript. There was no additional external funding received for this study.

**Competing interests:** The authors have declared that no competing interests exist.

## Introduction

The round goby *(Neogobius melanostomus)* is a benthic fish originally endemic to the Caspian and Black Sea that has successfully invaded numerous water bodies in North America, including the Great Lakes, and continues to expand its range [1]. Invaded ecosystems, including many that support important recreational and commercial fisheries, have exhibited dramatic changes and loss of native species [2, 3]. Round gobies predominantly inhabit shallow, rocky substrates, but can also be found in soft and sandy regions of freshwater and brackish environments [1, 4]. Round gobies not only compete with native fish for prey and habitat at young life stages, but also prey on a variety of fish eggs and fry, including those of lake trout (*Salvelinus namaycush*), lake sturgeon (*Acipenser fulvescens*), smallmouth bass (*Micropterus dolomieu*), and walleye (*Sander vitreus*) [1, 3, 5, 6]. However, round gobies are also consumed by the same fish species [7].

Although round gobies consume amphipods, chironomids, cladocerans, fish eggs and small fish [8–10], they feed heavily on dreissenid mussels, particularly after reaching sizes of 70 mm and larger [10–12]. Several studies have examined round goby diet using gut content analysis (GCA) and stable isotope analysis (SIA) [8, 10–13]. GCA has generally found that dreissenids are one of the main food sources for round gobies [10–13]. However, SIA has suggested that dreissenids are less important in the round goby diet than suggested by GCA, and non-shelled invertebrates such as chironomids and amphipods are more important [8, 9]. GCA may be biased because the method relies on visual identification of ingested food items and is therefore affected by differential digestion of various prey types, with shell fragments being persistent in guts. Further, GCA represents the most recent meal(s), so this approach may not be representative of the overall diet over a longer period without repeated sampling over time and space.

Compared to GCA, SIA provides an estimate of an organism's assimilated food over a period of weeks-months and may therefore be a better indicator than GCA of average diets [14]. However, SIA has its own biases. Stable isotope mixing models rely upon discrimination (sometimes called fractionation) values for the given tissue being analyzed, which are the difference between a consumer's and prey's isotopic values [15]. But discrimination factors can vary for many reasons, including taxa, growth rates, type of tissue, and digestion physiology which can all influence the rate-dependent kinetic reaction of isotope fractionation [16, 17]. The most used discrimination factors in aquatic food web studies are based on a meta-analysis by Post (2002) [15] and are 0.4‰ ± 1.3 for $\delta^{13}C$ and 3.4‰ ± 1.0 for $\delta^{15}N$ (mean ± SD). These values represent averages across a wide range of study organisms and show a relatively large range across various studies. If round goby discrimination factors differ from average values, this may explain inconsistencies between GCA and SIA results. We are not aware of any study determining discrimination factors specifically for round goby.

Here, we present results from an experiment designed to estimate discrimination factors for round gobies in both $\delta^{13}C$ and $\delta^{15}N$, and compare inference on round goby diets in Oneida Lake based on both the standard discrimination factors and our experiments. Round gobies were reared on four different prey types and muscle samples were collected for SIA to determine round goby-specific discrimination values. Round gobies were also collected from multiple locations in Oneida Lake to compare gut contents to stable isotope mixing model estimates. Based on prior studies, most round goby gut contents were expected to contain dreissenid mussels, and this proportion would increase as round goby size increased. We predicted that use of discrimination factors obtained specifically for round goby would provide better correspondence between GCA and SIA estimates of prey importance than the use of generalized values [15].

## Methods

### Stable isotope feeding study

Using a beach seine, 114 round gobies from 26–54 mm in length (mean = 40.6 mm) were collected from Oneida Lake (Latitude: 43.1737, Longitude: -75.9308; Fig 1) on 21 May 2019. Gobies were placed in fresh lake water before being transported to the experimental facility; total time from capture to introduction in the facility was <1 hour. Five to six round gobies each were kept in twenty-one 10-gallon aquariums (density: ~50 gobies/m$^2$). Sections of PVC pipe were added to each tank to provide cover and prevent aggression between individuals. Tanks were constantly aerated and ~25% of the water was changed twice weekly; room temperature was maintained to within 1˚C and was increased from 20˚C to 25˚C over the course of the study to mimic changes in Oneida Lake temperature. A 12:12 hour photoperiod was maintained by overhead fluorescent lighting. Ammonia levels were monitored daily with an API$^®$ Aquarium Ammonia Test Kit until all tanks reached undetectable levels (~3 weeks). The temperature in each tank was measured daily. When first introduced, the fish were not fed for 48 hours to limit ammonia build up. Eight fish died during the first week and were immediately removed from their tanks. One additional fish mortality occurred on day 26 of the experiment, otherwise no further mortality was observed. All fish handling followed Cornell's animal care protocol #2006–0088 and was approved by Cornell IACUC.

After the acclimation period of 48 h, round gobies were fed a single diet item: krill (San Francisco Bay Brand$^®$ Frozen Krill, *Euphausia superba*; n = 6 tanks), dreissenid mussels (quagga and zebra, picked by hand adjacent to where the gobies were collected and then frozen; n = 5 tanks), chironomid larvae (San Francisco Bay Brand$^®$ Frozen Bloodworms, *Chironomus plumosus*; n = 5 tanks), and Chilean mussels (PanaPesca$^®$ Cooked Mussel Meats, *Mytilus chilensis*; n = 5 tanks). Oneida Lake gobies consume mollusks and arthropods and these prey types were selected to represent both groups. Dreissenids and chironomids were used because they are common prey items of round goby [8–10]. Dreissenids were collected only from the area around the field station to minimize variability in isotopic signature of that prey item during the experiment, and were crushed open before being placed in the

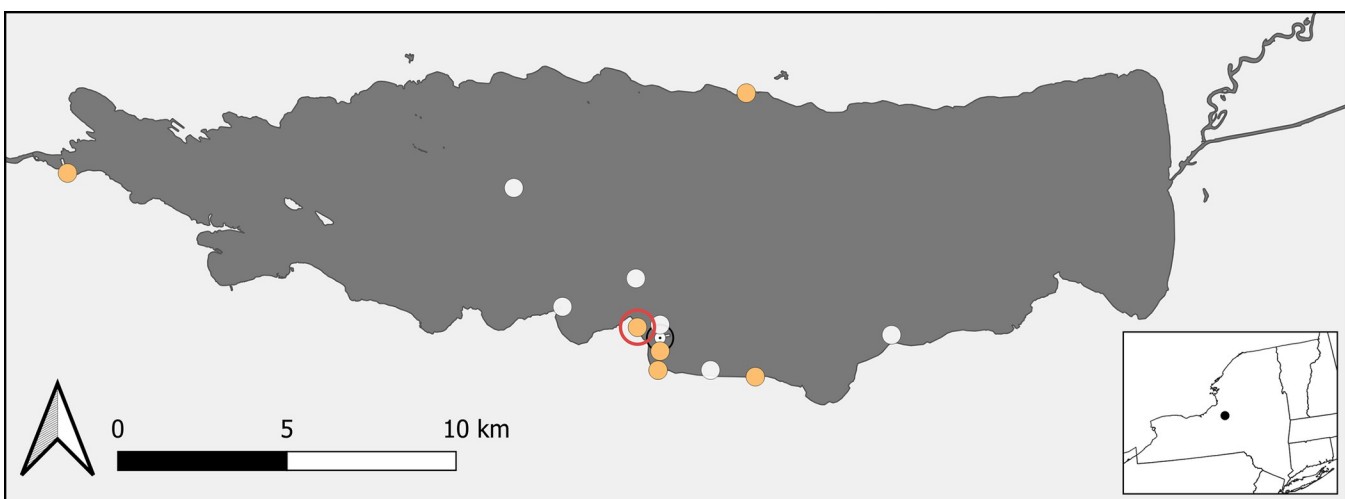

**Fig 1. Collection sites for round gobies in Oneida Lake, NY.** The hollow red circle surrounds the collection site for the round goby and dreissenid samples used in the feeding experiments. Orange circles mark collection sites of round goby for both stable isotope analyses and gut content analysis. White circles mark collection sites of round goby for gut content analysis. Two sites were close together and overlapped substantially as a result their location is marked by a white circle with a point inside of it.

experimental tanks. Chilean mussels and krill are marine species and have an isotopic signature deviating from that of the Oneida Lake round gobies, thereby allowing for better estimates of turnover rates and discrimination factors, by comparing isotopically distinct sources and maximizing the signal to noise ratio.

Round gobies were fed ad libitum daily, and unconsumed food was removed by hand after ~30 minutes. During the first two weeks of the feeding trial, a single individual food item (or piece of an individual food item) was placed in a microcentrifuge tube for SIA. Approximately every two weeks (on days 14, 31, 42, 55 and 63 of the experimental trial), three or four fish from each feeding group (12 or 16 fish total, and only one fish from each tank) were euthanized by submersion in buffered MS-222 (400 mg/L) for 15 minutes. Fish were weighed and their total length was measured to the nearest millimeter before dorsal white muscle samples were dissected for SIA.

### Stable isotope sample preparation and analysis

Muscle and food item samples were dried at ~60˚C for at least 48 hours, before being homogenized by grinding with a clean pestle in the microcentrifuge tubes. Subsamples of each (0.2–1.0 mg) were packed in tin capsules and analyzed for $\delta^{13}C$ and $\delta^{15}N$ using a Thermo Delta V isotope ratio mass spectrometer (IRMS) interfaced to a NC2500 elemental analyzer at the Cornell Isotope Laboratory (COIL). Every ten samples, an in-house standard was run to ensure accuracy and precision. Standard deviations for replicate analyses of standards were $\leq 0.07$ ‰ for $\delta^{13}C$ and $\leq 0.08$‰ for $\delta^{15}N$. Stable isotope values were expressed in the δ-notation in parts per mil according to the following equation:

$$\delta R\ (‰) = [(R_{sample} - R_{standard})/R_{standard}]\ x\ 1000 \tag{1}$$

where R is the ratio of heavy to light isotope, either $^{13}C/^{12}C$ or $^{15}N/^{14}N$ [18].

Based on C:N values, all food item $\delta^{13}C$ values required normalization for lipids (Table 1), while most round goby C:N values were low (range: 2.98–3.72) indicating that any lipid effects on $\delta^{13}C$ would be negligible, and therefore only C:N values >3.5 in fish samples were normalized for lipids [19]. We normalized $\delta^{13}C$ according to the following equation [19]:

$$Normalized\ \delta^{13}C = \delta^{13}C - 3.32 + 0.99\ x\ C:N \tag{2}$$

### Discrimination calculation

The change in muscle tissue $\delta^{13}C$ over time was estimated by fitting the asymptotic regression model adapted from [20] describing $\delta^{13}C$ changes in animals fed a prey with a known $\delta^{13}C$. This equation is:

$$\delta^{13}C_t = \delta^{13}C_e + (\delta^{13}C_e - \delta^{13}C_0) \cdot e^{(-\tau \cdot t)} \tag{3}$$

**Table 1. Isotope values (mean ± 1 SE) for all prey items in feeding experiment and average C:N for round gobies in the feeding experiment.**

|  | n | C:N | $\delta^{13}C$ | Normalized $\delta^{13}C$ | $\delta^{15}N$ |
|---|---|---|---|---|---|
| Chilean mussels | 4 | 5.25 ± 0.07 | -17.15 ± 0.58 | -15.28 ± 0.55 | 9.95 ± 0.36 |
| Chironomids | 6 | 4.84 ± 0.15 | -22.94 ± 0.42 | -21.47 ± 0.53 | 2.64 ± 1.42 |
| Dreissenids | 4 | 4.79 ± 0.11 | -26.91 ± 0.28 | -25.49 ± 0.34 | 8.31 ± 0.15 |
| Krill | 4 | 7.52 ± 1.62 | -29.50 ± 0.66 | -25.37 ± 1.38 | 3.63 ± 0.15 |
| Round gobies | 69 | 3.33 ± 0.21 | —— | —— | —— |

Normalized $\delta^{13}C$ are corrected for the effects of high lipid content (Eq 2).

where t is time in days, $\tau$ is the turnover rate in days, $\delta^{13}C_t$ is the C isotope ratio of gobies on day t, $\delta^{13}C_e$ is the isotope ratio of gobies when in equilibrium with their prey, and $\delta^{13}C_0$ is the isotope ratio of gobies on day 0. Because $\delta^{13}C_e$ equals $\delta^{13}C_{prey} + \Delta\delta^{13}C$, and $\delta^{13}C_0$, and $\delta^{13}C_{prey}$ are measured, we have an equation with two unknown parameters: the turnover rate $\tau$ and the discrimination factor $\Delta\delta^{13}C$. We assumed $\tau$ and $\Delta\delta^{13}C$ do not vary with prey type and used the non-linear fit function in Jmp 16.0 [21] to estimate $\tau$ and $\Delta\delta^{13}C$ and their approximate standard errors. The nonlinear fit function used the square deviation as the loss function and the analytic Gauss-Newton method. Each goby was treated as an independent replicate for a given date and diet as they were sampled from different aquarium for each sampling event. If the confidence limits of the estimate of $\Delta\delta^{13}C$ did not overlap with the standard value [15], we considered our values to be significantly different from the standard.

The process for estimating $\Delta\delta^{15}N$ and turnover rate for N is the same and the equation is identical to the equation for $\Delta\delta^{13}C$ after replacing $^{13}C$ with $^{15}N$. Turnover time was estimated separately for $\delta^{13}C$ and $\delta^{15}N$.

## Gut content sample collection and analysis

We collected 226 round gobies through a variety of sampling methods from twelve different sites within Oneida Lake (Fig 1). Fish were collected using a seine, trawl, or fishing rod between 0900 and 1500, late May-September, 2019. Upon capture, all round gobies were immediately placed into an overdose of buffered MS-222. Animals were also caught via electrofishing between 11PM– 12AM on 26 June 2019 and placed in fresh lake water before being euthanized by an overdose of buffered MS-222. Length (rounded to the nearest mm) and weight were recorded, and animals were frozen or placed in 95% ethanol within one hour of capture.

GCA was done by dissecting the entire digestive tract (round gobies do not have a true stomach; [8]), and then carefully opening the digestive tract. Contents were suspended in tap water and examined using a dissection scope at 10-40x magnification. All prey items were identified to the lowest classification practical [22] and enumerated. The length of all unfragmented dreissenids were measured to the nearest millimeter, and classified by species (zebra and quagga). Umbo pairs of fragmented and non-fragmented dreissenids were counted as one dreissenid mussel.

To estimate the importance of different diet items to round gobies, the percent frequency of occurrence for each diet item was calculated as follows:

$$\frac{\text{of fish with X prey item}}{\text{of fish with identifiable gut content}} \times 100\% \tag{4}$$

Frequency of occurrence provides a conservative estimate of prey importance as the total amount of a prey item in a stomach is not recorded [23]. Round gobies are gape limited; therefore, a linear regression was performed to determine whether a relationship existed between round goby total length and size of dreissenids consumed. Round gobies were also binned into four 25-mm size groups (17-42mm, 43-68mm, 69-94mm, and 95-120mm), as a size-dependent shift in diet has been observed to occur between 70 mm and 100 mm [8, 11]. The frequency of occurrence of the top three contributing prey items were calculated for each goby length group. Occasionally, an intact prey item (amphipod or chironomid) was collected from the gut and placed in a microcentrifuge tube for SIA.

## Isotope mixing models

Muscle samples from wild round gobies were used to compare how selection of discrimination factors influenced interpretation of round goby stable isotope ratios. Muscle samples were

collected from 45 freshly euthanized round gobies (size range = 33-117mm) caught in Oneida Lake. We used the Bayesian mixing model SIMMR [24] to estimate the contribution of dreissenids and other benthic organisms to these fish. Amphipods and chironomids were considered one end member (benthic) because they generally feed on detrital food sources. Dreissenids were considered one end member because they feed directly on pelagic phytoplankton [15]. Amphipods, chironomids, and dreissenids were collected from Oneida Lake.

The mixing models were evaluated twice to test the effect of discrimination factors on nutritional contributions. First, source contribution was estimated using the Post (2002) discrimination factors of 0.4‰ and 3.4‰ for $\Delta\delta^{13}C$ and $\Delta\delta^{15}N$, respectively [15]. The second model used discrimination values from our experiments, hereafter referred to as the experimental model. Round goby diets often shift as they increase in size [8, 11]; therefore, for both models, gobies were grouped by size based on their total lengths to best reflect sizes at which those diet shifts occur (<70mm: n = 20, 71-100mm: n = 14, >100mm: n = 11).

The models used Markov Chain Monte Carlo simulation methods (iterations = 10,000, burn in = 1000, thinning = 10, chains = 4). Convergence and fit were checked using Gelman diagnostics and plotting 50% posterior predictive distributions.

## Results

### Experiments

All round gobies were between 26 and 54 mm (total length, mean = 40.6, SD = 5.9) at the beginning of the experiment. Round gobies consumed all four prey types given and grew over the 63-day duration of the experiment to an average length of 45.3 mm (SD = 6.8) and from an average weight of 0.79 grams at the start to 1.13 grams on day 63 (S1 Fig). There was a clear progression towards the equilibrium stable isotope ratios in muscle tissue in all four feeding groups (Fig 2). The initial mean stable isotope ratios on day 0 of the experiment were -24.9‰ (SD = 2.6) for $\delta^{13}C$ and 14.4‰ (SD = 0.6) for $\delta^{15}N$.

The non-linear fit estimated the discrimination factors to be -0.4‰ for $\Delta\delta^{13}C$ and 4.0‰ for $\Delta\delta^{15}N$ with similar instantaneous turnover rates of ~0.02 (day$^{-1}$). The number of days until 50% muscle turnover was 30 and 36 days for $\delta^{13}C$ and $\delta^{15}N$, respectively (Table 2). The most informative prey item for calculating $\Delta\delta^{13}C$ and $\delta^{13}C$ turnover rate was Chilean mussels, which changed the $\delta^{13}C$ in gobies from -24.9‰ to -17.5‰ on day 63. The second most informative were the chironomids. Round goby $\delta^{13}C$ changed little (~1‰) over the course of the study in the other two feeding groups. For $\Delta\delta^{15}N$, the most informative prey were chironomids and krill. The model fit showed that the experimental $\Delta\delta^{13}C$ was lower than the "standard" value of 0.4‰, whereas the experimental $\Delta\delta^{15}N$ was higher than the standard value of 3.4‰.

### Gut content analysis

Of the 226 round gobies examined, 19 (8%) had empty guts, and 7 (3%) had only unidentifiable material within their digestive tracts. Chironomid larvae, dreissenid mussels and cladoceran zooplankton were the most frequently observed prey items (based on percent frequency occurrence); other benthic organisms (e.g., amphipods, snails, Trichoptera) were observed but in fewer round goby diets (Fig 3). The smallest size class of round goby (17–42 mm) mostly fed on cladocera and chironomids, while the largest size class (95–120 mm) predominantly fed on dreissenids (Fig 3). A clear transition from chironomid consumption to dreissenid consumption occurred in the intermediate size classes (43–68 mm and 69–94 mm), but fish in both size classes consumed a mix of both food sources (Fig 3).

Length of dreissenids found in guts increased with round goby length (slope = 5.07, t = 11.73, P <0.0001, n = 202), as did the minimum size of dreissenid consumed (Fig 4). The

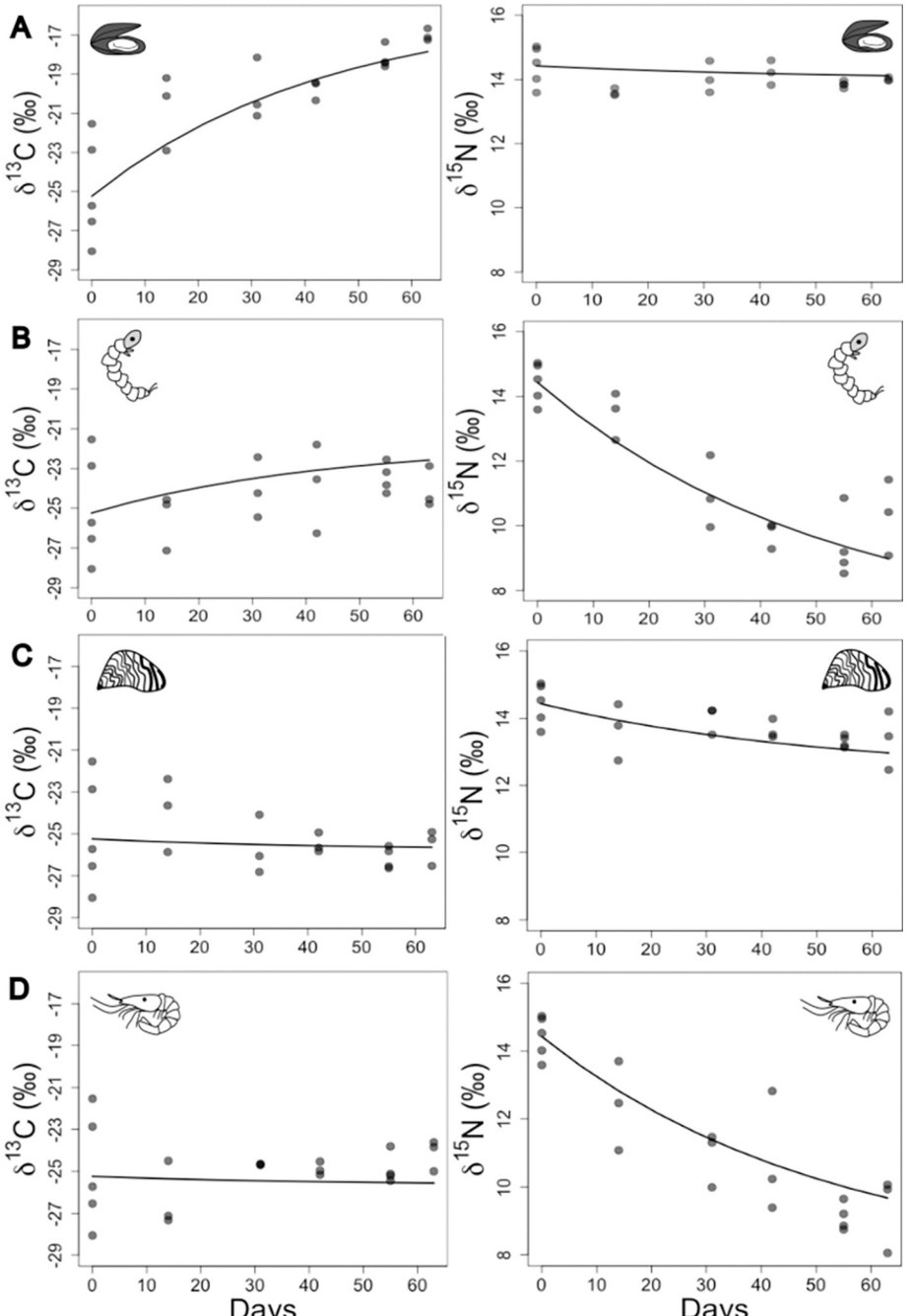

**Fig 2. Round goby stable isotope ratios by feeding group from all 63 days of the feeding trial.** (A = Chilean Mussels, *Mytilus chilensis*, B = Chironomids, *Chironomus plumosus*, C = Dreissenids, *Dreissena* spp., D = Krill, *Euphausia superba*). The line represents the predictions from the non-linear fit model used to determine discrimination factors and turnover rates.

largest mussels consumed were 12 mm and consumed by gobies >100 mm. More round gobies consumed zebra mussels (94% occurrence) than quagga mussels (35% occurrence) perhaps because the gobies were caught closer to shore where zebra mussels can be more abundant [25, 26].

**Table 2. Estimates from the non-linear fit to the experimental data.**

| Parameter | Estimate | Approximate SE | N | Root Mean Square Error |
|---|---|---|---|---|
| $\Delta\delta^{13}C$ | -0.41 | 0.32 | 64 | 1.31 |
| $\tau$ for C | 0.0229 | 0.0027 | | |
| $\Delta\delta^{15}N$ | 4.04 | 0.32 | 64 | 0.55 |
| $\tau$ for N | 0.0192 | 0.0020 | | |

## Stable isotope ratios and mixing models

Dreissenids from Oneida Lake, which were assumed to be supported by pelagic food sources, were ~3‰ more $\delta^{13}C$-depleted and ~1‰ more $\delta^{15}N$-depleted than benthic

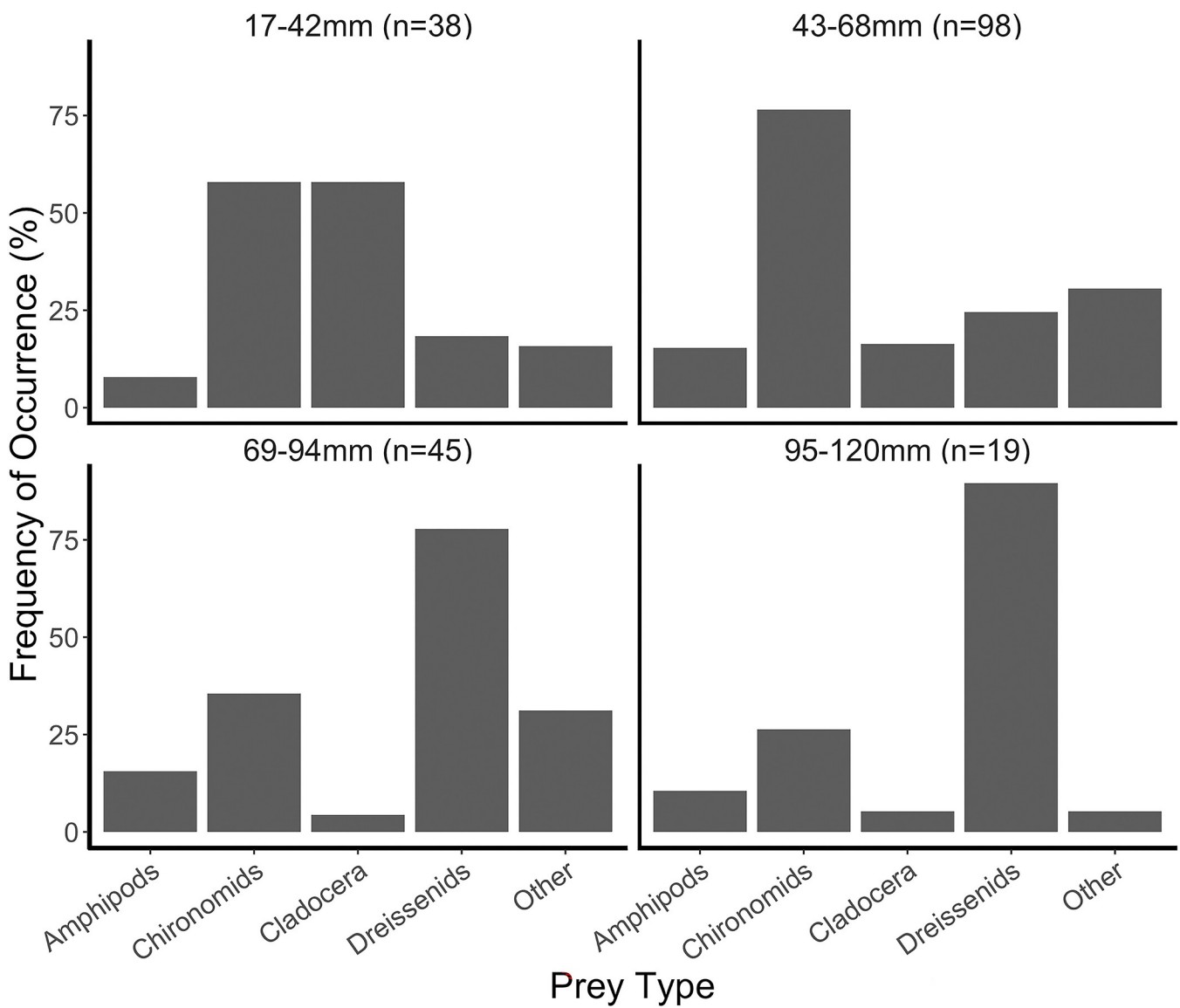

**Fig 3. Percent frequency of occurrence of prey items found in round goby guts, grouped into four 25-mm size bins, collected throughout Oneida Lake from May-September 2019.**

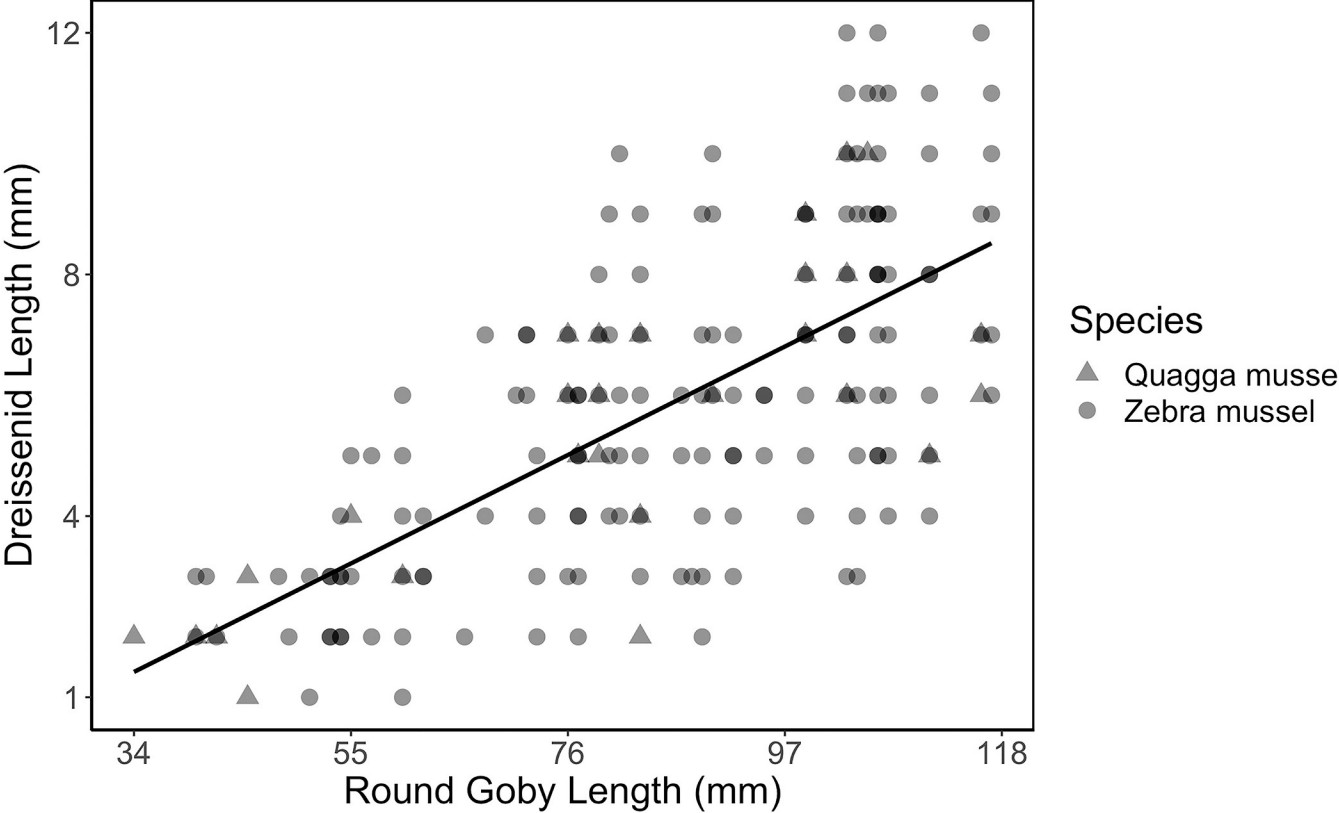

**Fig 4. Lengths of dreissenid mussels recovered from round goby guts plotted against the total length of the round gobies they were extracted from.** (Adj-$R^2$ = 0.46, p-value < 0.0001).

macroinvertebrates (Table 3). Round goby $\delta^{13}C$ values were negatively related to total length (df = 43, $R^2$ = 0.29, p < 0.001, Fig 5A), but there was no significant relationship between $\delta^{15}N$ and total length (df = 43, $R^2$ = 0.07, p = 0.08, Fig 5B). Instead, $\delta^{15}N$ was more variable in round gobies <70 mm than gobies >70 mm (Bartlett's K-squared: 8.74, p-value < 0.01; Fig 5B).

Both the standard and experimental models indicated pelagic pathways were the dominant contributor to round goby biomass across all size classes (median contributions ranging from 54–77%; Fig 6). The experimental model indicated higher contributions from pelagic pathways (dreissenids) in the largest fish size class (median contribution of 77%) compared to the standard model (median contributions of 59%). In the experimental model, benthic sources had a median contribution of 46% in round gobies 32-70mm (29–64%, 95% credible interval), 41% in mid-sized (71–100 mm) round gobies (22–61%), and 24% in round gobies 101-117mm (4–52%; Fig 6). The standard model estimated similar diet contributions across all size classes, with the highest pelagic pathway contribution in the mid-sized round goby class (median contribution of 68%).

**Table 3. Food source $\delta^{13}C$ and $\delta^{15}N$ values (mean ± 1 SD) used in isotope mixing models.**

|  | Benthic Members | | | Dreissenids | | |
|---|---|---|---|---|---|---|
|  | Mean | SD | N | Mean | SD | N |
| $\delta^{13}C$ | -22.3‰ | 2.2 | 3 | -25.5‰ | 0.7 | 4 |
| $\delta^{15}N$ | 9.4‰ | 0.8 | 3 | 8.3‰ | 0.3 | 4 |

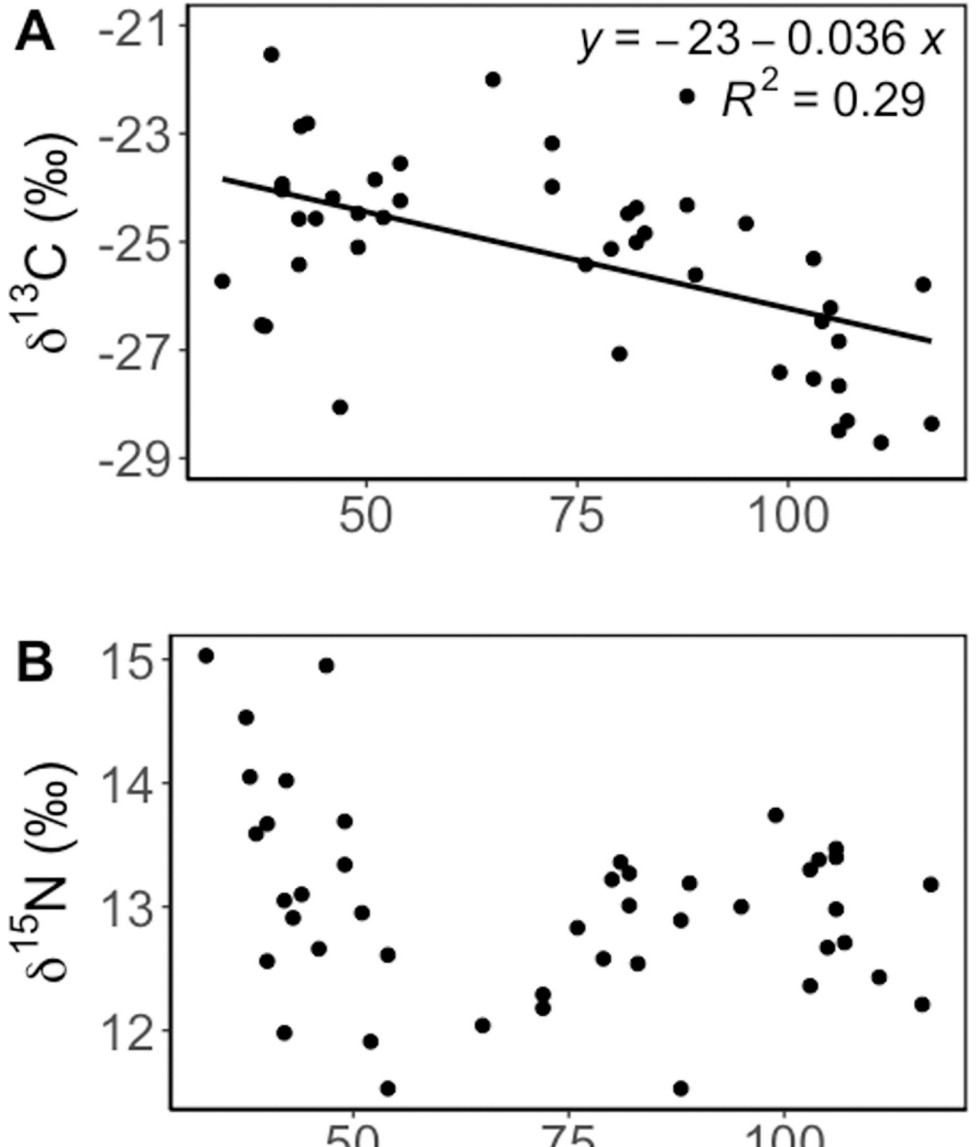

**Fig 5.** Relationship between round goby muscle tissue (A) $\delta^{13}$C and (B) $\delta^{15}$N and total animal length.

## Discussion

Trophic discrimination can vary widely in $\delta^{13}$C (~-3‰ to 4‰) and $\delta^{15}$N (~-1‰ to 6‰) [16, 17, 26] and can be related to the physiological state of the consumer as well as their diet [27]. Our experimental data resulted in discrimination factors that were 0.8‰ lower for $\Delta\delta^{13}$C and 0.6‰ higher for $\Delta\delta^{15}$N than the standard values presented by Post [15]. The standard values were outside the confidence limits from our experimental values for $\Delta\delta^{13}$C and just within the lower confidence limits for $\Delta\delta^{15}$N. The two mixing models had wide variation in credibility intervals (Fig 6), likely due to the benthic prey endmember group having a large $\delta^{13}$C standard deviation, and the two endmember groups having similar isotope values (Table 3; S3 & S4

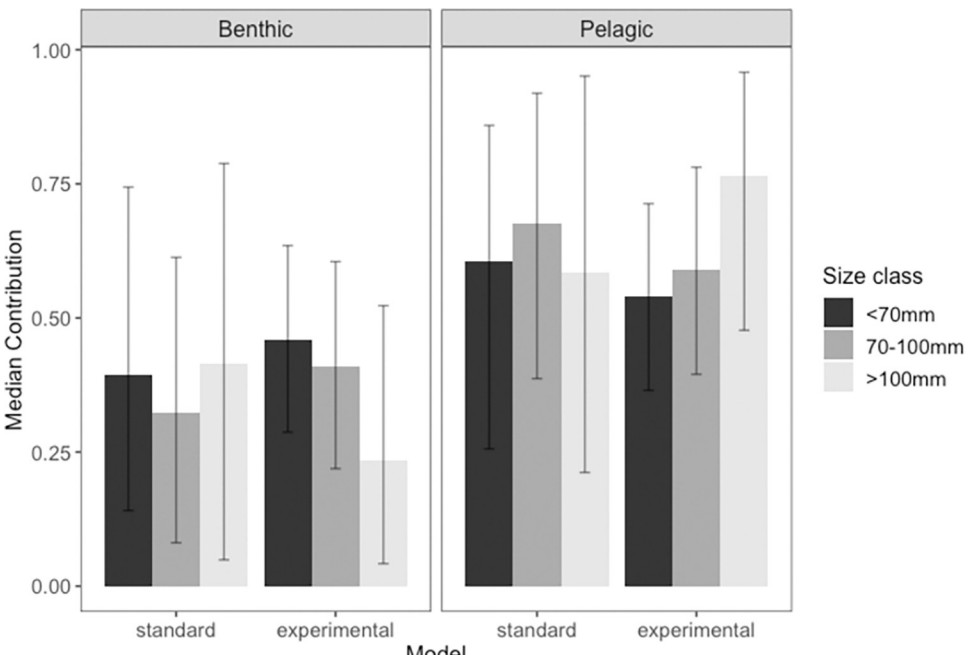

**Fig 6. Median diet contributions of benthic members and dreissenids for three round goby size classes.** Error bars indicate 95% credible intervals. Values calculated by two stable isotope analysis models with different discrimination factors (standard: commonly used discrimination values from Post [15], and experimental: discrimination values determined from the present study). See methods for a full description of how discrimination values were developed.

Figs). The mixing model utilizing the standard discrimination factors provided correspondence with GCA results, but the model using new discriminant factors agreed slightly better with GCA. With the new discriminant factors, SIA showed the same increasing trend of dreissenid use in larger gobies compared to smaller gobies, as seen in GCA. Therefore, disagreements in prior studies of dreissenid importance based on GCA or SIA [8, 28] can be partly explained by the choice of $\Delta\delta^{13}C$ and $\Delta\delta^{15}N$ values [8].

We are not aware of any other study on discrimination factors and isotopic turnover rates in round goby. Our $\Delta\delta^{13}C$ of -0.41‰ was slightly lower than literature values, which generally suggest mean values between 0‰ and 1‰ for dorsal white muscle tissues of fish [14, 15], with some studies even suggesting higher values of 1.5‰-3‰ [29, 30]. However, the range of values in the literature encompass our $\Delta\delta^{13}C$ value (mean ± 1 SD: 0.4 ± 1.3‰, [15]) and $\Delta\delta^{15}N$ value (3.4 ± 1.0‰, [15]). Our elevated $\Delta\delta^{15}N$ value of ~4‰ corresponds well with similar higher $\Delta\delta^{15}N$ (~4.5‰) obtained for cod (*Gadus morhua*) fed blue mussels (*Mytilus edulis*) [31]. Other studies have also found that predatory fish fed herbivorous invertebrates tended to have mean $\Delta\delta^{15}N$ values >3.4‰ (~3.8‰), although variability was high (~2‰) [32]. Our work reiterates that care and attention need to be taken when selecting discrimination values, and preferably validate the values selected for modeling with experiments.

The turnover rates derived from fitting Eq (3) to the experimental data were rapid, as expected for small fish growing quickly. The round gobies used in this experiment were small (~0.8 g; S1 Fig) and increased 43% in weight by the end of the experiment (~1.1 g; S1 Fig). The isotopic half-life in muscle tissue of similarly sized sand goby, *Pomatoschistus minutus*, (~1 g wet weight at the initiation of the experiment and ~2 g at end) were 25 and 28 days for $\delta^{13}C$ and $\delta^{15}N$, respectively [33]. More broadly, isotopic half-life in white muscle tissue from small fish is generally 25–60 days [33–35], and our model estimates indicated a similar rate of 30–36 days.

Wild-caught round gobies became 3‰ more $\delta^{13}$C-depleted with increasing size (Fig 5A), indicative of a shift towards assimilation of dreissenids in Oneida Lake, and a shift from benthic to pelagic diet pathways (Fig 6). Enriched $\delta^{13}$C signatures are a marker of benthic reliance due to carbon inputs from bacterial and meiofaunal sources, while depleted signatures indicate pelagic carbon inputs from phytoplankton. However, even some of the smallest round gobies had $\delta^{13}$C-depleted signatures consistent with pelagic pathways (Table 3, Fig 5). The smallest round goby may have a pelagic isotopic signature from feeding on cladocerans in addition to benthic prey, explaining the variability in isotope values in small fish. Zooplankton in Oneida lake are generally 2.7–5.4‰ more $\delta^{13}$C depleted than dreissenids [36]. The variability of $\delta^{15}$N was lower in larger animals (Fig 5B), consistent with a dietary specialization. Round gobies can reach lengths of 70mm by age 1, and >100mm by age 2 [1].

The benthic and dreissenid $\delta^{13}$C and $\delta^{15}$N values used in mixing models were similar to values reported in prior studies from Oneida Lake [36, 37]. In 2003, before the invasion of round goby, Bowman reported $\delta^{13}$C values in the summer of -26.8‰ and -24.5‰ for zebra mussel and amphipods respectively [36], which were ~1–2‰ more depleted than our $\delta^{13}$C values (-25.5 & -22.3). However, the benthic $\delta^{13}$C values in Oneida Lake are influenced by distance from shore, and our SIA sampling scheme was primarily in shallow, inshore waters (Fig 1), where aquatic animals are found to have less depleted $\delta^{13}$C values [37], while Bowman sampled offshore. The estimates for $\delta^{15}$N were also similar to prior studies [37, 38], which have found that the prey items we measured vary from ~8 to 10‰.

Round gobies function like many other benthic dwelling fish by consuming large numbers of benthic macroinvertebrates, but, unlike many small native benthic fish, are also able to utilize dreissenids. The smallest round goby observed to have consumed a dreissenid in the lake was 34 mm in length (Fig 4), however dreissenids did not become a major diet component until gobies reached >69 mm in total length (Fig 3). Small gobies consumed zooplankton, indicating that this group uses the open water more. In the midwater trawls conducted for GCA analysis, most of the round gobies caught (92%) were 22mm or less, although a very small number were caught up to 50mm. Small round gobies are regularly found in the open water in night fry trawls conducted by the Cornell Biological Field Station in Oneida Lake [38] and elsewhere [39]. Prior studies have also seen smaller round gobies consume mostly non-shelled benthic invertebrates and larger individuals increasingly depending on dreissenids [8, 13].

This study reinforces prior work that large round gobies are dreissenid specialists that also feed on other benthic prey sources [1]. In Oneida Lake, the arrival of round goby is correlated with, and probably the cause of, declines in several benthic invertebrate groups, including quagga mussels (zebra mussels declined before round goby arrived because of competition with quagga mussels) [7], as well as declines in small native benthic fish like darters [40]. Whether declines in the quagga mussels will lead to the lake ecosystem returning towards pre-mussel structure and function, as observed with zebra mussels [41], remains to be seen, but suggests that competition could increase with native benthic fishes if dreissenid populations decline.

Our experiments provide the first estimate of discrimination factors for round goby; factors that should be applicable to other systems. Although, a direct comparison between SIA and GCA is not possible because we did not measure percent volume of prey in gut contents, our experimentally determined discrimination factors resulted in a better correspondence between SIA and GCA and confirmed both the ontogenetic diet shift of round gobies and the high reliance of larger round gobies on dreissenid mussels in Oneida Lake. Stable isotope analysis provides complementary information to gut content when mixing models are properly parameterized.

## Supporting information

**S1 Fig. Weight (g) of sampled round gobies over 63-day feeding study (Adj-$R^2$ = 0.1641, p-value < 0.0001).**
(JPG)

**S2 Fig.** $\delta^{13}C$ and $\delta^{15}N$ of all feeding groups (A = Chilean mussels, B = chironomids, C = dreissenids, D = krill), with samples color coded by day of muscle sample collection. Squares indicate average isotope value for corresponding food items, while circles indicate round goby.
(PNG)

**S3 Fig. Isospace plot of experimental mixing model.**
(TIFF)

**S4 Fig. Isospace plot of standard mixing model.**
(TIFF)

**S1 Data.**
(XLSX)

## Acknowledgments

We thank staff and interns at Cornell Biological Field Station at Shackelton Point in the summer of 2019 for help and assistance during collection and preparation of samples.

## Author Contributions

**Conceptualization:** Anna M. Poslednik, Thomas M. Evans, Lars G. Rudstam.

**Data curation:** Anna M. Poslednik, Thomas M. Evans.

**Formal analysis:** Anna M. Poslednik, Thomas M. Evans, Lars G. Rudstam.

**Funding acquisition:** Thomas M. Evans, James R. Jackson, Lars G. Rudstam.

**Investigation:** Anna M. Poslednik, Thomas M. Evans.

**Methodology:** Thomas M. Evans, James R. Jackson, Lars G. Rudstam.

**Project administration:** Anna M. Poslednik, Thomas M. Evans.

**Resources:** Thomas M. Evans, James R. Jackson, Anthony J. VanDeValk, Thomas E. Brooking, Lars G. Rudstam.

**Software:** Anna M. Poslednik, Thomas M. Evans, Lars G. Rudstam.

**Validation:** Anna M. Poslednik, Thomas M. Evans, Lars G. Rudstam.

**Visualization:** Anna M. Poslednik.

**Writing – original draft:** Anna M. Poslednik.

**Writing – review & editing:** Anna M. Poslednik, Thomas M. Evans, James R. Jackson, Anthony J. VanDeValk, Thomas E. Brooking, Lars G. Rudstam.

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
