## [Decision Letter · Decision Letter 0]

29 Nov 2022

PONE-D-22-28011Round goby (*Neogobius melanostomus*) δ13C/δ15N discrimination values and comparisons of diets from gut content and stable isotopes in Oneida LakePLOS ONE

Dear Dr. Poslednik,

Thank you for submitting your manuscript to PLOS ONE. After careful consideration, we feel that it has merit but does not fully meet PLOS ONE’s publication criteria as it currently stands. Therefore, we invite you to submit a revised version of the manuscript that addresses the points raised during the review process.

We look forward to receiving your revised manuscript.

Kind regards,

Vitor Hugo Rodrigues Paiva, Ph.D.

Academic Editor

PLOS ONE

Journal Requirements:

"This research was supported by Cornell University, New York State Department of Environmental Conservation (NYSDEC) grant F-63-R to JRJ, and the John and Janet Forney endowment."

3. We note that Figure 1 in your submission contain map image which may be copyrighted. All PLOS content is published under the Creative Commons Attribution License (CC BY 4.0), which means that the manuscript, images, and Supporting Information files will be freely available online, and any third party is permitted to access, download, copy, distribute, and use these materials in any way, even commercially, with proper attribution. For these reasons, we cannot publish previously copyrighted maps or satellite images created using proprietary data, such as Google software (Google Maps, Street View, and Earth). For more information, see our copyright guidelines: http://journals.plos.org/plosone/s/licenses-and-copyright.

Reviewers' comments:

Reviewer's Responses to Questions

**Comments to the Author**

1. Is the manuscript technically sound, and do the data support the conclusions?

Reviewer #1: Yes

Reviewer #2: Yes

Reviewer #3: Partly

2. Has the statistical analysis been performed appropriately and rigorously? 

Reviewer #1: Yes

Reviewer #2: Yes

Reviewer #3: Yes

3. Have the authors made all data underlying the findings in their manuscript fully available?

Reviewer #1: Yes

Reviewer #2: Yes

Reviewer #3: No

4. Is the manuscript presented in an intelligible fashion and written in standard English?

Reviewer #1: Yes

Reviewer #2: Yes

Reviewer #3: Yes

5. Review Comments to the Author

Reviewer #1: This is a well-conducted study using experimental manipulation of diet to determine stable isotope discrimination factors for white muscle in round gobies, complemented with gut content and isotope measurements in wild fish. Overall the study was sound and the paper was very well written. My comments are meant to correct a few discrepancies, clarify a few word choices, and increase the generality of interpretations.

Line 75-76: This could be changed to be slightly clearer; as written it implies that the ‘high proportion’ of mussels will get more important as fish get bigger. It seems more accurate to say that mussels will get more important as fish get bigger. The wording is also unclear whether the proportion you refer to is within individuals (more mussels in each fish’s diet, which you did not measure) vs among individuals (more individuals have a mussel in their diet, which is what you did measure).

Line 206: It’s unclear why these numbers are different from those on line 83, as I thought both referred to the same fish.

Figure 2: The days on the x-axis don’t match with the methods, where you said samples were collected every 2 weeks. The data points are not evenly spaced (indicating samples were not taken at the same interval) and don’t match 14 days, including not ending on day 70 (10 weeks). Please correct the methods to reflect the true sampling protocol. I’d also suggest using smaller dots for each point, as there seems to be overlap among points that is hard to distinguish on the graph.

Line 223: Here you say the starting C was 25.4, but on line 211 you say starting C was 25.2. Also here, C values should be negative.

Line 226-7: While I agree for 7 of the 8 graphs, C for krill does not look like a good fit. Was ‘good fit’ a subjective determination?

Line 237 and paragraph: Remind the readers that this (I believe) is based on ‘percent frequency occurrence’ and so while it reflects how many fish consumed at least one of that particular prey item, it doesn’t show how much of each prey was consumed by individual fish (i.e. all fish eating one chironomid is not the same thing as all fish eating only chrionomids).

Line 242: I believe you mean “>” here, not <.

Figure 4: It would be more intuitive for the reader to change the axis units to reflect what was measured. So, for the y axis, only have whole numbers (since you rounded measurements to nearest mm) and make sure the min and max are included as numbers. Similarly for the x axis, make sure the min and max (or at least, much closer to those values) is included. Also include common names in the legend as that is what you use in the text. Were round goby lengths also rounded? It appears the data points are fairly regularly spaced along the x-axis, but rounded was not specified in the methods. Smaller dots would also help separate overlap, though if rounding was used and so dots completely overlap, consider using colour or size to indicate how many points overlap.

Figure 3: I’m unclear why ‘diptera’ is used in the figure yet referred to as ‘chironomid larvae’ throughout the text. Please be consistent. Also please define in text the y axis (gut content prevalence); is this the same as ‘percent frequency occurrence’ (line 178)? Or is it simply a count, in which case having counts also above each bar is unnecessary?

Line 239: The “other” category is second only to dreissenids for intermediate sized gobies, so perhaps should not be dismissed as ‘rare’.

Line 259-260: Why not test for differences in variation statistically? That would make conclusions based on your observations stronger, especially since this pattern seems to be driven by 3 individuals (looking at fig 5b) and so is not a particularly compelling point as written.

Line 260: Change 3B to 5B.

Line 287: What is the ‘interval state of the consumer’?

Line 308 and two paragraphs: Somewhere in here it seems relevant to include how long a round goby takes to grow in length. You reference weight under experimental conditions, but since your other analyses are based on length, that is more relevant, and especially line 324 talking about dietary switches, what can you tell us about how long an individual is within a size class?

Line 311: Please give scientific name.

Line 317: The graph doesn’t show us why the smallest are classified as pelagic; for that you need to also provide us with the numbers again (table 3), and tell the reader why C is a marker for pelagic vs benthic prey.

Line 318-320: This idea needs to be fleshed out more (and the wording changed, as they’re not ‘carbon’ depleted, it’s the isotope ratio that’s changed, not the amount of carbon). If I understand correctly, you’re saying that zooplankton (according to the reference you cite) are the most depleted in C, followed by dreissenids, then benthic prey. Wouldn’t your argument then predict small round gobies have the most depleted C, mid size gobies the highest C (when they use benthic prey), and then the largest gobies have intermediate C (when they switch to mussels?).

Line 321: Lower variability (which is still only driven by 3 individuals) indicates specialization but not on dreissenids per say; that is a different inference. There is also still very large variation amongst even the largest individuals, so conclusions of specialization (which I would say happens at size 50mm) should be made very cautiously.

Line 322-4: Information on goby growth rates would be helpful here.

Line 324-6: I don’t agree with this interpretation. Fig 2C shows that N values did decrease, and considering your table 3 shows that benthic and dreissenid N values were only 1 ‰ apart, and the decrease in the graph appears to be about 2 ‰, it seems the gobies did in fact change their N values in the dreissenid treatment.

Line 326-7: It’s unclear the point of this sentence – your data show that benthic food is still present in goby diets (fig 3), so are you saying they’ve eaten all the other benthic inverts, they’re competing with benthic inverts, there’s not enough benthic inverts for that to affect isotope values...?

Line 328: This paragraph is about temporal variation (and perhaps spatial) in C and N values. For the bigger picture, it would be useful to talk about why this is important for SIA studies. Even though you collected specimens from various locations, you could speak to inshore/offshore variation or south vs north shore variation (though it looks like deep water and north shore only have one sample each). How does this add variation to your results?

Line 330-1: The Bowman values are more depleted, not higher. It would also be easier on the reader to simply put your values in brackets next to the comparison values, instead of referring back to the table.

Line 340-341: If the small gobies cannot break open shells, do they get any nutrition from the ones they consume? You found gobies <50mm with shell fragments, so how is this physically possible? Since you argue above (lines 324-6) that experimental gobies on capture were already eating mussels, and these gobies were <54 or <50 (depending on whether line 206 or 83 is correct), doesn’t saying here that these small gobies can’t open shells contradict those statements?

Line 297 or perhaps a different paragraph: I think the discussion would benefit from additional discussions about the variability in isotope values especially for larger fish which you argue are specializing – this is not about discrimination values, but about what you can interpret ecologically from the values even with more accurate discrimination values. You have some very depleted C values (close to -29) – what does that say about some individual’s habitat use (assuming C reflect inshore/offshore gradient)? Were those the ones you collected in the deepest water? Similarly with the very high N values (15) of small fish – if pelagic pathway is 8.3 N, even with a discrimination factor of 4, these fish are still feeding at one higher trophic higher. How does that reconcile with your gut content results? I know you can’t directly compare them but you can certainly talk about discrepancies based on first principles of trophic enrichment and habitat use.

Reviewer #2: This study combined stomach content analysis, stable isotope analysis, and experimentally derived isotopic turnover rates to examine source contributions to the round goby and solve a discrepancy between diet studies which suggest gobies feed heavily on mussels and SIA which downplay the importance of mussels. In general, I found the manuscript to be well written with clear results and appropriate conclusions. Field studies examining the trophic structure of a single species are common but studies employing diet data, isotope data, and experimentally derived isotopic turnover rates are rare, and I commend the authors for combining all three methods.

My biggest point of contention is with the isotope mixing model. I think using a mixing model to assess the relative contributions of prey items makes sense but the isotopic similarities between the two prey endmember groups (benthic prey and mussels) is concerning as similar isotope values in sources can lead to problems with how the mixing model runs. Additionally, the sizeable error bars in model estimates of prey contributions and a lack of reporting of diagnostic tests used to assess model convergence and performance should be addressed. That said, those are still relatively minor changes and I believe once those issues (and a few others) are addressed, the manuscript will be ready for publication.

My comments, organized by major sections in the paper, are given below:

Figures:

- I like the inclusion of the study map but found the insert map of a partial view of New York to be unhelpful. Specifically, I think the insert would be more helpful if the view were a bit broader and zoomed out a little more.

- In Figure 1, gobies used for stomach content analysis were collected from a variety of points around the lake, but mussels were collected from a single location. Considering there can be spatial variation in the isotopic signatures of sessile filter feeders, providing some justification for the collection of mussels from a single location could be helpful.

- This is minor but in the text of the results the authors use weeks for their temporal scale but use days in Figure 2 and use weeks in one of the supplemental figures. Using a consistent time scale across results and figures would help the reader.

- In the methods, the primary metric used to assess diet was described as percent frequency of occurrence but “gut content prevalence” was used in Figure 3. Please consider changing one of the terms for consistency.

- Consider changing prey item labels in Figure 3 to match those used in text (mussels instead of Bivalvia, Chironomids instead of Diptera).

- Consider adding common names to Figure 4 to help people from other regions identify which mussel species is which.

Methods:

- Considering percent frequency of occurrence is one of several metrics used to assess the relative importance of prey items, I think it would be useful to provide some justification for only using %FO.

- Please report the results of diagnostic tests or metrics used to assess the performance/convergence of the mixing model.

Results:

- Line 223: Change to -25.4 and -17.5

- Line 243 is the first introduction of zebra vs quagga mussels so the sudden breakdown of how many gobies fed on each type of mussel comes a bit out of left field. Mentioning the two types of mussels earlier in the manuscript (methods?) would help alleviate any confusion.

- Line 260: change Fig 3B to Fig 5B

- The stable isotope ratios of the two end-member groups used in the SIMMR mixing model are similar in both their 13C and 15N ratios which could give the mixing model some trouble. Considering that it’s important for the baseline endmember isotope values to constrain the consumer isotope values, I think the authors should include an isospace plot showing the source and consumer data as a supplemental figure. An isospace plot, along with reporting the results of diagnostic tests used to assess model convergence would provide important context for readers familiar with isotope mixing models.

Discussion:

-The current study is restricted to a single lake within the Round Goby’s expansive introduced range. I would be interested in hearing the author’s thoughts on how applicable they think the findings of this study are to other regions where round gobies exist?

- There is considerable variation in the credibility intervals of diet contribution estimates made using SIMMR. Considering that variability, I would be interested to hear why the authors do or do not consider this to be a problem and what they think could be causing the wide variation in credibility intervals.

- Although the authors did a nice job of answering their primary question of what was causing the discrepancy in the estimated importance of mussels to round gobies when using SCA vs SIA, I think the authors miss a chance to discuss their findings in a broader context. Now that it has been demonstrated through multiple lines of evidence (SCA and SIA) that larger gobies feed heavily on Quagga and Zebra mussels, do these findings have important implications for the management of mussels or gobies? Do these findings change the way we view the ecological role of gobies or the way we view the ecosystems in which they live? The authors don’t need to address the above questions specifically, but these were the types of questions that came to mind when I finished the paper and was left wishing the results had been framed in a larger context during the discussion.

Reviewer #3: Line 83. Modify the opening phrase.

Line 87. Modify: … in twenty-one 10-gallon aquariums for in 21 aquariums (10 gallon), with five to six gobies per aquarium (density: ~50 gobies/m2)

Line 172. What identification keys were used?

Line 174. Change non-fragmented for unfragmented.

Line 176. It is better to use a different index to determine the most important item in the diet. I propose the use one of the following indexes:

1. Index of relative importance (IRI) proposed by Pinkas et al. 1971

2. Prey-specific index of relative importance (PSIRI) proposed by Brown et al. 2012.

Because if you only use the percent frequency of occurrence this value could be bias due that the weight of the preys is not taking in account.

- Brown SC, Bizzarro JJ, Cailliet GM, Ebert DA (2012) Breaking with tradition: redefining measures for diet description with a case study of the Aleutian skate Bathyraja aleutica (Gilbert 1896). Environ Biol Fish 95:3–20

- Pinkas L, Oliphant MS, Iverson ILK (1971) Food habits of albacore, bluefin tuna and bonito in California waters. Fish Bull Calif Dep Fish Game 152:47–63

Lines 163 and 233. There is a mismatch between number of samples for gut content analysis… in methods you said 226, but in results you mentioned 225… which is the correct number?

Lines 180-182. How did you choose this scale for split the samples in three size groups? Because you have more length (mm) in the second and third group… it is not comparable. I mean: first group: 35-17 = 18 mm; second: 70-35 = 34 mm; third: 117-71 = 46 mm. Better if you use Sturges rule or if you have a biological condition like the shift in diet, made some intervals that have the same length; e.g., 17 – 42; 43 – 68; 69 – 94; 95- 120, all the ranges have 25 mm; and in the third group you have the group that represent the shift in diet.

Line 187. Isotope mixing models. Why don’t you used isotopic niche analysis? Whit this you also can plot the both sources that you are evaluated and determine how is the distribution of both sources and consumer in a biplot (d13C and d15N).

Lines 234-240. You describe the diet that the most commonly consumed prey items were, Chironomid larvae, dreissenid mussels and cladoceran zooplankton but when the figure 3 was reviewed, the items were classified as the order that they belong. So, it is better if the description in the text (results) are the same in the figure, therefore I suggest, to modified the text or modified the figure for a concordance between the manes of the prey items.

Lines 240-242. Rephrase the sentences.

Figure 4. Could you please include the tendency line of the model.

Lines 243-245. Again, could you please include in the text the scientific names of zebra and quagga mussels, because in the Fig. 4, you just have the scientific names.

Line 247. I suggest to modified the place where the Fig. 3 is going to be. I propose that you split the paragraph which start in the line 233 and finish in line 245, so you can include the Fig. 3 before you talk about the relation between fish and mussels. It will be better if the figure 3 will be in line 242.

Lines 287-291. Redundant sentence, says the same as the previous sentence.

Line 293. Explain how discriminant factors can affect the GCA results?

Line 316. More than a shift towards assimilation of dreissenids is the fact that they possible change from benthic to pelagic… you can also see this in fig. 6 with the experimental results.

Lines 324-327. I don’t think that this not change in Fig. 2C suggest that the gobies were already feeding on dreissenids when captured, you have tendency but the variance between all points is high.

And the last phrase of this paragraph it has no relation to what is mentioned before.

Lines 338-341. You started the phrase with “The smallest round goby observed to have consumed a dreissenid was 34 mm in length”, but then you mentioned that the “Round gobies smaller than 50 mm lack the pharyngeal teeth strength required to break open dreissenid shells” … so how do you explain that small individuals (34 mm), could feed on those mussels?

- In the reviewed bibliography for comparing your results with previously works on round goby, you miss the work of:

Skabeikis Artūras, Lesutienė Jūratė .2015. Feeding activity and diet composition of round goby (Neogobius melanostomus, Pallas 1814) in the coastal waters of SE Baltic Sea. International Journal of Oceanography and Hydrobiology 44(4): 508-519.

- I do not know If is just due that the document you upload to the platform modified the quality of the figures, but all are really poor quality. If possible, improve the quality.

6. PLOS authors have the option to publish the peer review history of their article (what does this mean?). If published, this will include your full peer review and any attached files.

Reviewer #1: No

Reviewer #2: No

Reviewer #3: No

---

## [Author Response · Author response to Decision Letter 0]

10 Feb 2023

Please see "Response to Reviewers" letter, which has been copied and pasted below for your convenience:

We thank the editor for the opportunity to edit and resubmit the manuscript. We carefully considered and attempted to address each of the reviewer’s comments. 

Figure 1 has been remade in QGIS, which is a free and open-source software and should comply with the CC BY 4.0 license.

Reviewer #1:

Line 75-76: This could be changed to be slightly clearer; as written it implies that the ‘high proportion’ of mussels will get more important as fish get bigger. It seems more accurate to say that mussels will get more important as fish get bigger. The wording is also unclear whether the proportion you refer to is within individuals (more mussels in each fish’s diet, which you did not measure) vs among individuals (more individuals have a mussel in their diet, which is what you did measure). Accepted following the reviewer's suggestion; wording for clarity lines 75-76. 

Line 206: It’s unclear why these numbers are different from those on line 83, as I thought both referred to the same fish. Line 83 was changed to reflect the true numbers in line 206 (rather than rounded numbers). 

Figure 2: The days on the x-axis don’t match with the methods, where you said samples were collected every 2 weeks. The data points are not evenly spaced (indicating samples were not taken at the same interval) and don’t match 14 days, including not ending on day 70 (10 weeks). Please correct the methods to reflect the true sampling protocol. I’d also suggest using smaller dots for each point, as there seems to be overlap among points that is hard to distinguish on the graph. We have edited figure 2 to have smaller dots and changed x-axis to reflect sampling days. Edited methods to include days of sampling (line 123). 

Line 223: Here you say the starting C was 25.4, but on line 211 you say starting C was 25.2. Also here, C values should be negative. Value should have been -25.2, this has been changed now (line 226). 

Line 226-7: While I agree for 7 of the 8 graphs, C for krill does not look like a good fit. Was ‘good fit’ a subjective determination? The sentence was deleted (line 240). 

Line 237 and paragraph: Remind the readers that this (I believe) is based on ‘percent frequency occurrence’ and so while it reflects how many fish consumed at least one of that particular prey item, it doesn’t show how much of each prey was consumed by individual fish (i.e. all fish eating one chironomid is not the same thing as all fish eating only chrionomids). We have now included the phrase “based on percent frequency occurrence” in lines 249-250.

Line 242: I believe you mean “>” here, not <. Accepted following the reviewer's suggestion (line 262). 

Figure 4: It would be more intuitive for the reader to change the axis units to reflect what was measured. So, for the y axis, only have whole numbers (since you rounded measurements to nearest mm) and make sure the min and max are included as numbers. Similarly for the x axis, make sure the min and max (or at least, much closer to those values) is included. Also include common names in the legend as that is what you use in the text. Were round goby lengths also rounded? It appears the data points are fairly regularly spaced along the x-axis, but rounded was not specified in the methods. Smaller dots would also help separate overlap, though if rounding was used and so dots completely overlap, consider using colour or size to indicate how many points overlap. We edited figure 4 to include all suggestions. Included the clause “rounded to nearest mm” in methods, line 176.

Figure 3: I’m unclear why ‘diptera’ is used in the figure yet referred to as ‘chironomid larvae’ throughout the text. Please be consistent. Also please define in text the y axis (gut content prevalence); is this the same as ‘percent frequency occurrence’ (line 178)? Or is it simply a count, in which case having counts also above each bar is unnecessary? We edited figure 3 to have consistent prey names with manuscript, and changed y-axis to say: “frequency of occurrence (%)”. 

Line 239: The “other” category is second only to dreissenids for intermediate sized gobies, so perhaps should not be dismissed as ‘rare’. Accepted following the reviewer's suggestion, changed to “in fewer” diets (line 251). 

Line 259-260: Why not test for differences in variation statistically? That would make conclusions based on your observations stronger, especially since this pattern seems to be driven by 3 individuals (looking at fig 5b) and so is not a particularly compelling point as written. Accepted following the reviewer's suggestion. We have now tested using Bartlett’s K-squared and included results in line 276. 

Line 260: Change 3B to 5B. Accepted following the reviewer's suggestion (line 276). 

Line 287: What is the ‘interval state of the consumer’? The ‘interval state’ was the result of a misspelling and has been changed to “the physiological state of the consumer,” line 303. 

Line 308 and two paragraphs: Somewhere in here it seems relevant to include how long a round goby takes to grow in length. You reference weight under experimental conditions, but since your other analyses are based on length, that is more relevant, and especially line 324 talking about dietary switches, what can you tell us about how long an individual is within a size class? We have now included a sentence on round goby growth, lines 342-343. 

Line 311: Please give scientific name. Accepted following the reviewer's suggestion, line 329.

Line 317: The graph doesn’t show us why the smallest are classified as pelagic; for that you need to also provide us with the numbers again (table 3), and tell the reader why C is a marker for pelagic vs benthic prey. We have now included this information in table 3 in parentheses (line 338) and included a sentence on why C is a marker for pelagic vs benthic (lines 335-337).

Line 318-320: This idea needs to be fleshed out more (and the wording changed, as they’re not ‘carbon’ depleted, it’s the isotope ratio that’s changed, not the amount of carbon). If I understand correctly, you’re saying that zooplankton (according to the reference you cite) are the most depleted in C, followed by dreissenids, then benthic prey. Wouldn’t your argument then predict small round gobies have the most depleted C, mid size gobies the highest C (when they use benthic prey), and then the largest gobies have intermediate C (when they switch to mussels?). We have now discussed the variability in our C values in small fish, and this is because they eat a mix of cladocera and benthic prey, not just cladocera. We also edited the paragraph to more clearly express this in lines 343-346.

Line 321: Lower variability (which is still only driven by 3 individuals) indicates specialization but not on dreissenids per say; that is a different inference. There is also still very large variation amongst even the largest individuals, so conclusions of specialization (which I would say happens at size 50mm) should be made very cautiously. We have deleted “towards dreissenids” in line 342. 

Line 322-4: Information on goby growth rates would be helpful here. We have included a sentence on round goby growth, lines 342-343. 

Line 324-6: I don’t agree with this interpretation. Fig 2C shows that N values did decrease, and considering your table 3 shows that benthic and dreissenid N values were only 1 ‰ apart, and the decrease in the graph appears to be about 2 ‰, it seems the gobies did in fact change their N values in the dreissenid treatment. We have deleted from line 346.

Line 326-7: It’s unclear the point of this sentence – your data show that benthic food is still present in goby diets (fig 3), so are you saying they’ve eaten all the other benthic inverts, they’re competing with benthic inverts, there’s not enough benthic inverts for that to affect isotope values...? This has been deleted from line 346.

Line 328: This paragraph is about temporal variation (and perhaps spatial) in C and N values. For the bigger picture, it would be useful to talk about why this is important for SIA studies. Even though you collected specimens from various locations, you could speak to inshore/offshore variation or south vs north shore variation (though it looks like deep water and north shore only have one sample each). How does this add variation to your results? All SIA samples were collected in shallow, inshore waters, and therefore we cannot speak to variation between locations, included this clause in lines 352-353.

Line 330-1: The Bowman values are more depleted, not higher. It would also be easier on the reader to simply put your values in brackets next to the comparison values, instead of referring back to the table. We have this to “more depleted” and included values in parentheses (line 350). 

Line 340-341: If the small gobies cannot break open shells, do they get any nutrition from the ones they consume? You found gobies <50mm with shell fragments, so how is this physically possible? Since you argue above (lines 324-6) that experimental gobies on capture were already eating mussels, and these gobies were <54 or <50 (depending on whether line 206 or 83 is correct), doesn’t saying here that these small gobies can’t open shells contradict those statements? Swallowing small dreissenids whole, without being able to crack the shell will likely result in very little nutritional uptake (lines 360-363).

Line 297 or perhaps a different paragraph: I think the discussion would benefit from additional discussions about the variability in isotope values especially for larger fish which you argue are specializing – this is not about discrimination values, but about what you can interpret ecologically from the values even with more accurate discrimination values. You have some very depleted C values (close to -29) – what does that say about some individual’s habitat use (assuming C reflect inshore/offshore gradient)? Were those the ones you collected in the deepest water? Similarly with the very high N values (15) of small fish – if pelagic pathway is 8.3 N, even with a discrimination factor of 4, these fish are still feeding at one higher trophic higher. How does that reconcile with your gut content results? I know you can’t directly compare them but you can certainly talk about discrepancies based on first principles of trophic enrichment and habitat use. This paragraph (lines 302-314) has been rewritten and modified to better draw the gut content and stable isotope data together for comparison before considering the mixing models.

Reviewer #2:

My biggest point of contention is with the isotope mixing model. I think using a mixing model to assess the relative contributions of prey items makes sense but the isotopic similarities between the two prey endmember groups (benthic prey and mussels) is concerning as similar isotope values in sources can lead to problems with how the mixing model runs. Additionally, the sizeable error bars in model estimates of prey contributions and a lack of reporting of diagnostic tests used to assess model convergence and performance should be addressed. That said, those are still relatively minor changes and I believe once those issues (and a few others) are addressed, the manuscript will be ready for publication. We have reported diagnostic tests used to assess model convergence and performance in the methods sections (lines 215-217) and included a discussion on the similar isotope values between endmembers (lines 307-310). 

Figures:

- I like the inclusion of the study map but found the insert map of a partial view of New York to be unhelpful. Specifically, I think the insert would be more helpful if the view were a bit broader and zoomed out a little more. Figure 1 has been edited following the reviewer’s comments.

- In Figure 1, gobies used for stomach content analysis were collected from a variety of points around the lake, but mussels were collected from a single location. Considering there can be spatial variation in the isotopic signatures of sessile filter feeders, providing some justification for the collection of mussels from a single location could be helpful. The justification of minimizing variability for mixing models included in lines 114-116.

- This is minor but in the text of the results the authors use weeks for their temporal scale but use days in Figure 2 and use weeks in one of the supplemental figures. Using a consistent time scale across results and figures would help the reader. The manuscript has been edited throughout to discuss the experiment in terms of days, relevant figures have axes in days now.

- In the methods, the primary metric used to assess diet was described as percent frequency of occurrence but “gut content prevalence” was used in Figure 3. Please consider changing one of the terms for consistency. The figure 3 y-axis is now labeled “Frequency of Occurrence (%)”.

- Consider changing prey item labels in Figure 3 to match those used in text (mussels instead of Bivalvia, Chironomids instead of Diptera). Figure 3 prey items labels now match those used in text. 

- Consider adding common names to Figure 4 to help people from other regions identify which mussel species is which. Figure 4 legend now has common names.

Methods:

- Considering percent frequency of occurrence is one of several metrics used to assess the relative importance of prey items, I think it would be useful to provide some justification for only using %FO. We have added a justification in lines 189-190.

- Please report the results of diagnostic tests or metrics used to assess the performance/convergence of the mixing model. Accepted following the reviewer's suggestion. These are now included in lines 215-217.

Results:

- Line 223: Change to -25.4 and -17.5 Accepted following the reviewer’s suggestion (line 238).

- Line 243 is the first introduction of zebra vs quagga mussels so the sudden breakdown of how many gobies fed on each type of mussel comes a bit out of left field. Mentioning the two types of mussels earlier in the manuscript (methods?) would help alleviate any confusion. Accepted following the reviewer's suggestion, included in lines 183-184.

- Line 260: change Fig 3B to Fig 5B Accepted following the reviewer's suggestion (line 276).

- The stable isotope ratios of the two end-member groups used in the SIMMR mixing model are similar in both their 13C and 15N ratios which could give the mixing model some trouble. Considering that it’s important for the baseline endmember isotope values to constrain the consumer isotope values, I think the authors should include an isospace plot showing the source and consumer data as a supplemental figure. An isospace plot, along with reporting the results of diagnostic tests used to assess model convergence would provide important context for readers familiar with isotope mixing models. These are now included as supplemental figures 3 & 4. 

Discussion:

-The current study is restricted to a single lake within the Round Goby’s expansive introduced range. I would be interested in hearing the author’s thoughts on how applicable they think the findings of this study are to other regions where round gobies exist? We have included a clause in line 379 about how the discrimination factors determined for round goby should be easily applicable to other systems. 

- There is considerable variation in the credibility intervals of diet contribution estimates made using SIMMR. Considering that variability, I would be interested to hear why the authors do or do not consider this to be a problem and what they think could be causing the wide variation in credibility intervals. The wide credible intervals are due to the two prey endmember groups being so close in isotope value, which does not allow us to state significant difference between the two models. We included a statement on this and edited some phrasing in lines 307-310.

- Although the authors did a nice job of answering their primary question of what was causing the discrepancy in the estimated importance of mussels to round gobies when using SCA vs SIA, I think the authors miss a chance to discuss their findings in a broader context. Now that it has been demonstrated through multiple lines of evidence (SCA and SIA) that larger gobies feed heavily on Quagga and Zebra mussels, do these findings have important implications for the management of mussels or gobies? Do these findings change the way we view the ecological role of gobies or the way we view the ecosystems in which they live? The authors don’t need to address the above questions specifically, but these were the types of questions that came to mind when I finished the paper and was left wishing the results had been framed in a larger context during the discussion. Included a paragraph addressing some broader implications in lines 370-377.

Reviewer #3: 

Line 83. Modify the opening phrase. Accepted following the reviewer's suggestion (line 83).

Line 87. Modify: … in twenty-one 10-gallon aquariums for in 21 aquariums (10 gallon), with five to six gobies per aquarium (density: ~50 gobies/m2) Accepted following the reviewer's suggestion (lines 86-87).

Line 172. What identification keys were used? Accepted following the reviewer's suggestion; source is now cited as [21].

Line 174. Change non-fragmented for unfragmented. Accepted following the reviewer's suggestion (line 183).

Line 176. It is better to use a different index to determine the most important item in the diet. I propose the use one of the following indexes:

1. Index of relative importance (IRI) proposed by Pinkas et al. 1971

2. Prey-specific index of relative importance (PSIRI) proposed by Brown et al. 2012.

Because if you only use the percent frequency of occurrence this value could be bias due that the weight of the preys is not taking in account.

- Brown SC, Bizzarro JJ, Cailliet GM, Ebert DA (2012) Breaking with tradition: redefining measures for diet description with a case study of the Aleutian skate Bathyraja aleutica (Gilbert 1896). Environ Biol Fish 95:3–20

- Pinkas L, Oliphant MS, Iverson ILK (1971) Food habits of albacore, bluefin tuna and bonito in California waters. Fish Bull Calif Dep Fish Game 152:47–63

We did not measure volume/weight of gut content or prey items recovered from guts, and we cannot use any of these other indexes. We did include a justification for %FO in the text now (lines 189-190). 

Lines 163 and 233. There is a mismatch between number of samples for gut content analysis… in methods you said 226, but in results you mentioned 225… which is the correct number? The correct number is 226. This has been fixed in manuscript. 

Lines 180-182. How did you choose this scale for split the samples in three size groups? Because you have more length (mm) in the second and third group… it is not comparable. I mean: first group: 35-17 = 18 mm; second: 70-35 = 34 mm; third: 117-71 = 46 mm. Better if you use Sturges rule or if you have a biological condition like the shift in diet, made some intervals that have the same length; e.g., 17 – 42; 43 – 68; 69 – 94; 95- 120, all the ranges have 25 mm; and in the third group you have the group that represent the shift in diet. Accepted following the reviewer's suggestion at line 193 and in Figure 3.

Line 187. Isotope mixing models. Why don’t you used isotopic niche analysis? Whit this you also can plot the both sources that you are evaluated and determine how is the distribution of both sources and consumer in a biplot (d13C and d15N). Isotopic niche analysis is better suited for studies incorporating multiple species to better understand community structure and sizes of niches, while in this study we examine the isotopic signature and diet of a single species of round goby. We have included two isospace figures (one from each mixing model) as supplemental figures to better demonstrate the distributions of sources and consumers (S3 & S4). 

Lines 234-240. You describe the diet that the most commonly consumed prey items were, Chironomid larvae, dreissenid mussels and cladoceran zooplankton but when the figure 3 was reviewed, the items were classified as the order that they belong. So, it is better if the description in the text (results) are the same in the figure, therefore I suggest, to modified the text or modified the figure for a concordance between the manes of the prey items. We have modified figure 3 to match prey names in text.

Lines 240-242. Rephrase the sentences. Accepted following the reviewer's suggestion (line 260).

Figure 4. Could you please include the tendency line of the model. Accepted following the reviewer's suggestion, included trend line in Figure 4.

Lines 243-245. Again, could you please include in the text the scientific names of zebra and quagga mussels, because in the Fig. 4, you just have the scientific names. We have modified Figure 4 to include common names. 

Line 247. I suggest to modified the place where the Fig. 3 is going to be. I propose that you split the paragraph which start in the line 233 and finish in line 245, so you can include the Fig. 3 before you talk about the relation between fish and mussels. It will be better if the figure 3 will be in line 242. The paragraph was split into two (lines 247-265).

Lines 287-291. Redundant sentence, says the same as the previous sentence. We have deleted this sentence (line 305).

Line 293. Explain how discriminant factors can affect the GCA results? We have worked to make the phrasing clearer (lines 311-312).

Line 316. More than a shift towards assimilation of dreissenids is the fact that they possible change from benthic to pelagic… you can also see this in fig. 6 with the experimental results. We have included this in a clause in lines 334-335.

Lines 324-327. I don’t think that this not change in Fig. 2C suggest that the gobies were already feeding on dreissenids when captured, you have tendency but the variance between all points is high. Accepted following the reviewer's suggestion (deleted in line 346).

And the last phrase of this paragraph it has no relation to what is mentioned before. Accepted following the reviewer's suggestion (deleted in line 346).

Lines 338-341. You started the phrase with “The smallest round goby observed to have consumed a dreissenid was 34 mm in length”, but then you mentioned that the “Round gobies smaller than 50 mm lack the pharyngeal teeth strength required to break open dreissenid shells” … so how do you explain that small individuals (34 mm), could feed on those mussels? We have included a statement on how small round gobies swallow can small dreissenids whole, but are likely not acquiring many nutrients (lines 360-363).

- In the reviewed bibliography for comparing your results with previously works on round goby, you miss the work of:

Skabeikis Artūras, Lesutienė Jūratė .2015. Feeding activity and diet composition of round goby (Neogobius melanostomus, Pallas 1814) in the coastal waters of SE Baltic Sea. International Journal of Oceanography and Hydrobiology 44(4): 508-519. This has now been incorporated and is included as citation [39].

- I do not know If is just due that the document you upload to the platform modified the quality of the figures, but all are really poor quality. If possible, improve the quality. Remade all figures, hopefully better quality now.

---

## [Decision Letter · Decision Letter 1]

7 Mar 2023

PONE-D-22-28011R1

Round goby (*Neogobius melanostomus*) δ13C/δ15N discrimination values and comparisons of diets from gut content and stable isotopes in Oneida Lake

PLOS ONE

Dear Dr. Poslednik,

Thank you for submitting your manuscript to PLOS ONE. After careful consideration, we feel that it has merit but does not fully meet PLOS ONE’s publication criteria as it currently stands. Therefore, we invite you to submit a revised version of the manuscript that addresses the points raised during the review process.

We look forward to receiving your revised manuscript.

Kind regards,

Vitor Hugo Rodrigues Paiva, Ph.D.

Academic Editor

PLOS ONE

Reviewers' comments:

Reviewer's Responses to Questions

**Comments to the Author**

1. If the authors have adequately addressed your comments raised in a previous round of review and you feel that this manuscript is now acceptable for publication, you may indicate that here to bypass the “Comments to the Author” section, enter your conflict of interest statement in the “Confidential to Editor” section, and submit your "Accept" recommendation.

Reviewer #1: All comments have been addressed

Reviewer #3: All comments have been addressed

2. Is the manuscript technically sound, and do the data support the conclusions?

Reviewer #1: (No Response)

Reviewer #3: Partly

3. Has the statistical analysis been performed appropriately and rigorously? 

Reviewer #1: (No Response)

Reviewer #3: No

4. Have the authors made all data underlying the findings in their manuscript fully available?

Reviewer #1: (No Response)

Reviewer #3: No

5. Is the manuscript presented in an intelligible fashion and written in standard English?

Reviewer #1: (No Response)

Reviewer #3: Yes

6. Review Comments to the Author

Reviewer #1: Though the authors have sufficiently addressed my comments, I have made a few more which will only serve to improve the quality of the ms if they are addressed (they will improve clarity and consistency).

Line 41: I don’t think that an emphasis on ‘sport fish’ is necessary; importance of effects shouldn't be determined by fish we 'care' about because we can fish for them.

Line 55-56: This critique is true if you only have one study at one time point, but becomes less of an argument if you have multiple studies (which you reference) that have collected diet at different time points.

Line 58: ‘average’ is a better descriptor than ‘overall’. GCA works better for rare items; SIA captures the contribution of either the dominant food or averages the effects of many foods.

Line 61-2: Here it would be good to actually discuss the reasons why discrimination factors differ, as they are the reason you argue you need specific factors for gobies.

Line 76: This is a prediction, not a hypothesis.

Line 124: I believe ‘or’ instead of ‘to’ is more accurate word choice.

Line 234: Please use a consistent number of significant digits throughout (eg., these are different from the abstract)

Line 234 and 241: In the first line, you present the C discrimination factor as negative, but in the second line you did not. Please be consistent and in line with the literature.

Table 2: Why does Δδ 15N have a tau but Δδ13C does not?

Line 302: Add ‰ to ‘-3’.

Line 330: Please review the ms to make sure you are referring to δ and not absolute levels of C and N (i.e. make sure to use the delta notation when you are referring to changes in isotope ratios, as you are here, and not changes in the amount of carbon or nitrogen, which you are not measuring).

Line 344-5: Saying ‘shifting between the two sources’ suggests as they get larger, some are now specializing on benthic prey, which I believe is not what you intended you say.

Line 345-6: But didn’t you sample gobies across several months? Are you suggesting that period just happens to be when they are all switching between a mixed diet to a pelagic (mussel) diet? I’m not really sure what you’re arguing in the latter half of this paragraph (which trend you’re trying to explain); is it ontogenetic diet shifts? Variability?

Line 349: Specify these are deltaC values.

Line 350: Again, please make sure you’re always providing the appropriate units and specifying what the numbers are.

Line 349-353: Unclear comparison – if your value for amphipods is -22.3, why are you comparing that to perch values?

Paragraph line 347: Perhaps give more of a context for this paragraph – why is temporal and spatial variability in isotope values important?

Line 362-3: So when you said earlier you measured shell fragments, does that mean the shells were intact in these smaller gobies? Wouldn’t that be important information to give from the GCA? It’s also unclear why these small gobies were consume items that they were not getting nutrition from – isn’t that a low-fitness strategy? Why would that behavioural trait persist?

Paragraph line 370: This paragraph is unclear. Round gobies invaded after zebra and quagga mussels? Native fish weren't declining before gobies were introduced? Are both quagga and zebra mussels declining in the lake? Are you implying the round gobies are responsible for the decline in quagga mussels? You last sentence is also just a thought, it seems, with no support; sure that's possible, but it's also possible they'll do fine together (how do gobies coexist with species in their native range?)

Reviewer #3: The authors made all the recommendation previous send, but I think there is extra subjects that need to be addressed.

Line 19. Remove “over the 63 days”, you already mentioned in line 17.

Line 49. Include the reference [39]

Line 52. Include the information of the reference 39. If is the same that this works reports previously.

Line 63. Change, Post from 2002 by Post (2002) [14].

Line 83. Include, 114 round gobies “from” 26-54 mm in length…

Line 114. Reference.

Lines 117-119. How the fact that some prey items that are different from the area where the gobies were collected, allowing a better estimate of turnover rates and discrimination factor.

How can you deal with the fact that you have some item preys that were from Oneida Lake and some outside this area?

Line 132. How much sample weight for SIA?

Line 141. If you have values above 3.5, you need to normalize the data, as that is the set threshold. You mention that the range goes up to 3.72, those values need standardization.

Lines 189-190. Even though you include a “justification” for only used %FO, this justification should be with a reference that support the statement, if not, it is just an opinion, not a justification.

Line 190. What do you mean with “Round gobies are gape limited; therefore” it is talk about gape mouth? Or what gape?

Line 209. Instead of used “the “standard” discrimination factors” … used the Post (2002) discrimination factors…

Line 238. You mentioned before that the most informative prey item was Chilean mussels (Line 236-238). But then in line 238 said: “The next most informative were the chironomids“ which one is the correct?

Table 3. You mentioned in lines 204 – 207: Amphipods and chironomids were considered one end member (benthic) because they generally feed on detrital food sources. Dreissenids were considered one end member because they feed directly on pelagic phytoplankton [14]. Amphipods, chironomids, and dreissenids were collected from Oneida Lake.

But when the table 3 appears, you only have 3 (carbon) and 4 (nitrogen) individuals for each, benthic and dreissenids groups. So, for benthic I want to know how many amphipods and chrinomids you have in the group, and it is a really a concern that the SD in the d13C values for benthic group is higher than 2%... it is a really really high variation… so, I don’t know if is due to the sample size, or the fact that you mix to different taxa or both.

Lines 307-308. What do you mean with “credibility intervals”?

Additionally, …“due to the two prey endmember groups having have similar isotopic values” … this is not true. With a difference higher than 2% do you think that the groups have the same values? For this statement be true you should have done a statistical analysis, otherwise it is just an opinion with no mathematical support.

Lines 361-363. You started the phrase with “The smallest round goby observed to have consumed a dreissenid was 34 mm in length”, but then you mentioned that the “Round gobies smaller than 50 mm lack the pharyngeal teeth strength required to break open dreissenid shells” … so how do you explain that small individuals (34 mm), could feed on those mussels? We have included a statement on how small round gobies swallow can small dreissenids whole, but are likely not acquiring many nutrients (lines 360-363).

Ok, but if this food item does not provide a large amount of nutrients, and also cannot be opened to extract the individual... what is the explanation for that small gobies feeding of it?

Mixing models and isotopic niche. What do you mean with pelagic and bentic in both isotopic niche graphics? It is better if you include all the food items A=Chilean mussels, B=chironomids, C=dreissenids, D=krill), and all the round goby isotopic information. Otherwise, pelagic and bentic, it doesn't tell me anything, there is no reference to what that is this. If you combine all the food items and the goby you could have a better idea about the behavior of the values, with this I mean, who far or close are the food items (prey) with respect of the goby (predators). Supplementary material S3 and S4.

If you create the better graphs you should mentioned the figures in the text.

7. PLOS authors have the option to publish the peer review history of their article (what does this mean?). If published, this will include your full peer review and any attached files.

Reviewer #1: No

Reviewer #3: **Yes: **Tatiana A. Acosta-Pachon

---

## [Author Response · Author response to Decision Letter 1]

7 Apr 2023

We carefully considered and attempted to address each of the reviewers' comments in our revised manuscript. Please see attached "Response to Reviewers" document for detailed comments.

---

## [Editor Report · Decision Letter 2]

12 Apr 2023

Round goby (*Neogobius melanostomus*) δ13C/δ15N discrimination values and comparisons of diets from gut content and stable isotopes in Oneida Lake

PONE-D-22-28011R2

Dear Dr. Poslednik,

We’re pleased to inform you that your manuscript has been judged scientifically suitable for publication and will be formally accepted for publication once it meets all outstanding technical requirements.

Kind regards,

Vitor Hugo Rodrigues Paiva, Ph.D.

Academic Editor

PLOS ONE
---

## [Editor Report · Acceptance letter]

14 Apr 2023

PONE-D-22-28011R2 

Round goby (*Neogobius melanostomus*) δ13C/δ15N discrimination values and comparisons of diets from gut content and stable isotopes in Oneida Lake 

Dear Dr. Poslednik:

I'm pleased to inform you that your manuscript has been deemed suitable for publication in PLOS ONE. Congratulations! Your manuscript is now with our production department. 

Kind regards, 

on behalf of

Dr. Vitor Hugo Rodrigues Paiva 

Academic Editor

PLOS ONE